# The Resonance and Adaptation of *Neurospora crassa* Circadian and Conidiation Rhyth ms to Short Light-Dark Cycles

**DOI:** 10.3390/jof8010027

**Published:** 2021-12-29

**Authors:** Huan Ma, Luyao Li, Jie Yan, Yin Zhang, Xiaohong Ma, Yunzhen Li, Yu Yuan, Xiaolin Yang, Ling Yang, Jinhu Guo

**Affiliations:** 1Key Laboratory of Gene Engineering of the Ministry of Education, State Key Laboratory of Biocontrol, School of Life Sciences, Sun Yat-sen University, Guangzhou 510006, China; mhdx001@126.com (H.M.); liluyao@mail2.sysu.edu.cn (L.L.); 61642921.hi@163.com (Y.Z.); maxh25@mail2.sysu.edu.cn (X.M.); liyzh27@mail2.sysu.edu.cn (Y.L.); yuay227@mail.sysu.edu.cn (Y.Y.); yangxlin3@mail2.sysu.edu.cn (X.Y.); 2School of Life Sciences, Jiaying University, Meizhou 514015, China; 3Center for Systems Biology, School of Mathematical Sciences, Soochow University, Suzhou 215006, China; yanjie@suda.edu.cn (J.Y.); lyang@suda.edu.cn (L.Y.)

**Keywords:** *Neurospora crassa*, circadian clock, FREQUENCY (FRQ), conidiation, light/dark cycle

## Abstract

Circadian clocks control the physiological and behavioral rhythms to adapt to the environment with a period of ~24 h. However, the influences and mechanisms of the extreme light/dark cycles on the circadian clock remain unclear. We showed that, in *Neurospora crassa*, both the growth and the microconidia production contribute to adaptation in LD12:12 (12 h light/12 h dark, periodically). Mathematical modeling and experiments demonstrate that in short LD cycles, the expression of the core clock protein FREQUENCY was entrained to the LD cycles when LD > 3:3 while it free ran when *T* ≤ LD3:3. The conidial rhythmicity can resonate with a series of different LD conditions. Moreover, we demonstrate that the existence of unknown blue light photoreceptor(s) and the circadian clock might promote the conidiation rhythms that resonate with the environment. The ubiquitin E3 ligase FWD-1 and the previously described CRY-dependent oscillator system were implicated in regulating conidiation under short LD conditions. These findings shed new light on the resonance of *Neurospora* circadian clock and conidiation rhythms to short LD cycles, which may benefit the understandings of both the basic regulatory aspects of circadian clock and the adaptation of physiological rhythms to the extreme conditions.

## 1. Introduction

The Earth rotates with a period of approximately 24 h, which causes periodic changes in many environmental factors. Circadian clocks are the inner mechanisms that allow organisms to adjust their physiology and behavior to the daily cycling of environmental factors, e.g., light and temperature [1].

In the natural daily cycling conditions, rhythms with the same 24 h period to the environmental factors occur in organisms, which are called diurnal rhythms. Whereas in constant conditions in which any environmental cues are absent, e.g., light, dark, temperature or humidity, organisms display rhythms with periods slightly deviated from 24 h, which are called free-running rhythms [2]. The period length of regular alternation of temporal cues, e.g., light and dark, which is different from 24 h, is defined as *T*-cycle [3]. In certain ranges, non-24 h light-dark (LD) cycles can entrain the circadian rhythms, and the flexibility of this entrainability dramatically varies between different organisms. Within the range of entrainment, which is also called the limit of stable entrainment, the rhythm changes in accordance with the non-24 h *T*-cycles and show stable phase angles. Phase angle represents the relationship between the timing of the rhythm and the timing of an external time cue, which can be measured by selecting a point in the entraining cycle (e.g., light on), and a phase reference point in the resulting rhythm (e.g., onset, trough or peak of the rhythm). Whereas beyond the ranges, the circadian clocks free-run [4,5,6,7,8].

The entrainment ranges have been characterized in a number of species including human, hamster and orange bread mold (*Neurospora crassa*) [9,10,11,12,13,14,15]. In the wild-type filamentous fungus *N**. crassa*, the conidiation banding rhythms are consistent with a range of *T*-cycles that include LD6:6, LD12:12 and LD14:14, which illustrates the masking effects [14]. These observations have been extended to conidiation rhythms in LD3:3 and LD9:9 [11]. However, under extremely short *T*-cycles that exceed the thresholds, the free running rhythms have been observed in some species, including *Neurospora*, Jack bean (*Canavalia ensiformis*) and sparrow (*Passer domesticus*) [9,12].

It is very challenging for the endogenous circadian rhythms to be entrained to non-24 h LD cycles compared to the rest-activity rhythms [10,16]. Under such conditions, the endogenous rhythmicities may desynchronize with the behavioral rhythmicities, which leads to inadaptation or disorders that have been observed in a number of checked organisms [10,17,18,19].

*Neurospora* is an important model organism for circadian clock study, as it exhibits an overt clock phenotype in its conidiation. On the molecular level, the core circadian clock system of *Neurospora* is composed of two positive elements: White Collar 1 (WC-1) and WC-2 and one negative element: FREQUENCY (FRQ). The FRQ/WC-based circadian oscillator (FWO) dictates the circadian rhythms at molecular and physiological levels [20,21]. WC-1 harbors three PER-ARNT-SIM (PAS) domains; it is responsible for light responses of the circadian clock as a blue light photoreceptor [22,23]. WC-1 and WC-2 form the heterodimer white collar complex (WCC) to activate the transcription of *frq*, and the translated protein FRQ acts as negative a component to repress the function of WCC, which constitute the negative feedback loop in regulating the circadian expression of the clock genes and clock-controlled genes [1].

*Neurospora* has several additional validated or putative photosensing-associated factors, including VVD, CRY, NOP-1, PHY-1 and PHY-2 [24,25,26,27,28,29,30]. VIVID (VVD), another PAS domain-containing protein, is a flavin-binding blue-light photoreceptor that regulates the light responses and temperature compensation of the *Neurospora* circadian clock [27,30,31,32,33,34]. Interestingly, a cry-dependent oscillator gate-1 (*cog-1*) mutation led to conidiation rhythmicity in constant light in *Neurospora*, suggesting a role for cog-1 in regulating light sensing in the circadian clock. The *cog-1* related oscillator is called the CRY-dependent oscillator (CDO) system since it requires the blue-light photoreceptor CRYPTOCHROME (CRY) [14]. Cryptochrome is a putative blue light sensor, and its function remains elusive [27,35]. PHY1/2 are putative red-light receptors and NOP-1 is a putative green light receptor [36].

In this work, we investigated the influence of short light-dark cycling conditions on the circadian clock in *Neurospora*, and compared the effects of the different light-dark cycles on growth, microconidia production, conidiation rhythmicity and gene expression. The findings of this work would shed new light on the knowledge of circadian clocks under conditions with extreme *T*-cycles.

## 2. Materials and Methods

### 2.1. Media, Growth Conditions and Transformation Procedure

The *Neurospora crassa 301-5* (*ras-1^bd^*, *a*) strain, obtained from Dr. Yi Liu’s lab, was used as the wild-type strain in this work [37]. The 2% LCM media contained 1 × Vogel’s medium with 0.17% arginine and 2% glucose [38]. The race tube solid media consisted of 1 × Vogel’s medium containing 2% glucose, 50 ng of biotin/mL, and 1.5% agar.

The *315-13* (*fwd-1^RIP^*, *his-3*) strain, a strain in which the *fwd-1* was disrupted, was obtained from Dr Yi Liu’s lab and described previously [39]. The *vvd^KO^* strain was obtained from FGSC [40]. All of the strains information are listed in Appendix A.

Conidia were inoculated to seed mycelial mats in petri dishes, and disks were cut from these and used for 50-mL liquid cultures under certain LD conditions. At designated time points the cultures were harvested by filtration, frozen in liquid nitrogen, and ground in liquid nitrogen.

An incubator (Percival Scientific, USA) was used to simulate different LD cycles, the white light intensity during light was 5000 lux or 1000 lux as indicated. The red light (λ = 660 nm) was generated by an LED lamp, and the intensity was 5000 lux or 1000 lux as indicated. The blue light (λ = 470 nm) was generated by an LED lamp, and the intensity was 5000 lux or 1000 lux as indicated.

The counting of microconidia was conducted on a hemocytometer under an optical microscope. Briefly, since the shape of microconidia is nearly round and they usually contain a single nucleus, we consider those hyphae with length/width ratio <1.5 as a subjective criteria as microconidia. The proportion of microconidia was the ratio of the microconidia quantity divided by the total quantity of both microconidia and macroconidia.

### 2.2. Race Tube Assay

The rhythmicity of *Neurospora* conidiation is observed in race tubes, which are hollow glass growth tubes, bent up at both ends in order to contain the solid agar medium sufficient for linear growth for ~one week. In a race tube assay, the conidia are inoculated on the surface of the solid media at one end of the race tube thus it grows toward the other end. During growth, *Neurospora* yields conidiation bands periodically, which permit the measurement of the period length by calculating the growth time between two adjacent conidiation bands [21].

In this work, the conidiation banding profiles were assayed on race tubes under standard conditions [14]. The growth front was marked every 24 h under a red safe light in all light conditions in this work. All race tube experiments were carried out at room temperature (25 °C).

### 2.3. Protein Analysis

Protein extraction, quantification and western blot analysis were conducted as described elsewhere [41,42]. For Western blotting, equal amounts of total protein (50 μg) were loaded in each protein lane of SDS-PAGE (7.5%) gels containing a ratio of 37.5:1 acrylamide/bisacrylamide.

### 2.4. Dynamic Modeling

The modeling was conducted based on the following assumptions: (1) *frq* mRNA is synthesized in the nucleus and transfers to the cytosol where it is degraded; (2) nuclear FRQ protein, and combined the processes of multiple phosphorylation and translocation of the nuclear to a time delay; (3) the rate of synthesis of FRQ is proportional to *frq* mRNA several h ago; (4) *wc-1* mRNA and two states of nuclear WCC complex are involved in this model: phosphorylated and light dependent; (5) to consider the adaptation effect to light, VVD is introduced to the model (Appendix A).

### 2.5. RNA-Seq and Analysis

The strains subjected to RNA-seq were grown and the RNA samples were isolated from duplicates. All of the Pearson’s correlation values between each duplicate were >0.99. Tophat was used as the aligner to map the reads to the reference genome [*N. crassa* OR74A (NC12)]. The genes showing ≥1.5-fold increase were selected for further analysis. See detailed RNA-seq methods and parameters in Appendix A.

### 2.6. Statistical Analysis

Data are mean ± SE or mean ± SD where indicated. *n* ≥ 3. The Student’s t test was used for all statistical analyses. * Represents the *p*-value of the statistical tests is less than the significance level of 0.05 (*p* ≤ 0.05); ** represents *p* ≤ 0.01 and *** represents *p* ≤ 0.001.

## 3. Results

### 3.1. Subsections

#### 3.1.1. The Response and Resonance of *Neurospora* Conidiation Rhythms under LD Routines

To address how the short LD cycles affect the rhythms, we conducted race tube assays with the wild-type strain (*301-5, bd*), under the following series of short LD cycling conditions including LD9:9, LD6:6, LD4:4, LD3:3, LD2:2, LD1:1, LD45 min:45 min and LD30 min:30 min, which consisted of symmetric light and dark phases. Additionally, we tested *Neurospora* growth in LD65 min:25 min, an asymmetric condition, to mimic the light/dark condition on orbital flight. In orbital flight, the light–dark cycling period is about 90 min in which 2/3 in light and 1/3 in dark [43,44]. The results show that under LD9:9, LD6:6, LD4:4, LD3:3, LD2:2, the conidiation displayed overt rhythms with periods that coincided with the LD cycles. However, when the cycles were shorter (<LD2:2), the entrained conidiation rhythms were absent (Figure 1A).

The arrhythmicity of *Neurospora* under extremely short LD cycles (e.g., LD1:1) could be explained by two possibilities: (1) the conidiation rhythms were abolished; (2) the bands were too compact and ambiguous to detect. To validate, we used the non-band strain FGSC4200 which grows very fast in race tube. In constant dark, FGSC4200 shows mild conidiation rhythms [45], while in LD1:1 it showed weak but recognizable conidiation rhythms (Appendix A). These data suggest that although the free-running circadian period of *Neurospora* is approximately 22 h [20], the conidiation rhythm can be driven by extremely short LD cycles.

The circadian clock can adjust to fit with the cycling environment, especially the light. We compared the growth rate of two *frq* mutants *frq^2^* and *frq^7^*, which possess shorter and longer free-running periods in constant dark, respectively [46]. The race tube assay results reveal that in LD9:9 the growth ratio of *frq^2^* vs. *frq^7^* was higher than that in LD13.5:13.5 (Figure 1B,C; Appendix A), suggesting a resonance of *Neurospora* circadian period to the environment.

However, when we compared the growth rate of the wild-type strain under various LD cycles, we obtained unexpected results showing that *Neurospora* grew much faster in constant dark than that in constant light or most LD conditions. Moreover, the growth rate in shorter LD cycles was not necessarily slower than LD12:12. For instance, the growth in constant darkness (DD), LD6:6, LD3:3 and LD1:1 was significantly faster than that in LD12:12. The growth under LD65 min:25 min was also significantly faster than LD12:12 (Figure 1D). The dramatic differences in growth under LL vs. DD, LD45 min:45 min vs. LD65 min:25 min indicate that the growth is influenced by the daily length of lighting in *Neurospora*.

*Neurospora* microconidia are small uninucleate spores that serve as male gametes or as asexual reproductive structures which are important to stress adaptation [47]. When we counted the microconidia produced under LD12:12 and short LD cycles, respectively, and the results show that the proportion of microconidia in LD12:12 was significantly higher than those in other LD conditions. Statistic results revealed a significant correlation between the LD cycles and microconidia proportion while there was no significant correlation between LD cycles and growth rates (Figure 1E and Appendix A).

#### 3.1.2. Expression of the FRQ Protein in Different LD Cycles

As the circadian clock is involved in controlling the conidiation rhythmicity, we asked whether the molecular rhythmicity of the FRQ protein is subject to changes in response to the length of the LD cycles. As one of the core factors in *Neurospora* circadian clock, FRQ acts as the negative component in the transcription/translational feedback loop [20]. We conducted mathematical modeling and the results suggested that under LD12:12 and LD6:6, the concentration of FRQ protein matches the LD cycles (Figure 2A; Appendix A; protocol and parameters for modeling in Appendix A), instead, under LD2:2 the FRQ levels show a free running pattern with some saw-tooth shaped fluctuations. The simulation results show that the mRNA levels have a direct response to light, but the protein curves are smoother, especially for high-frequency light signals. Therefore, the post-translational regulatory and translocation processes may perform a buffering function or high-frequency filtering function. Furthermore, the numerical data also predicted that the direct response of *frq* mRNA to the light stimuli is lower under shorter LD cycles (Figure 2A), which may be due to the light adaptation module (VVD-WCC loop) and the core negative feedback loop.

We next conducted Western blotting and showed that under LD12:12 and LD6:6, the expression of the FRQ protein and the phosphorylation profile displayed coincident rhythms with the periods of LD cycles. However, under LD3:3 and shorter cycles, the FRQ levels seemed to free run but we also observed some saw-tooth shaped fluctuations under LD3:3 (Figure 2B–G). These data support the mathematical modeling.

#### 3.1.3. FWO Is Not Required for Adaptation of Conidiation Rhythms to Short LD Cycles in High-Intensity Light

In the *Neurospora* circadian oscillator, WC-1 and WC-2 are the two positive elements in which WC-1 functions as a blue light sensor [23]. We asked whether WCC proteins are responsible for conidiation rhythmicity under LD cycles, therefore, we grew the *wc-1^RIP^*, *wc-2^KO^* and *wcc^DKO^* and *frq^10^* strains in race tubes under different LD cycles and observed condition rhythms for all of these mutants exhibit conidiation rhythms under LD12:12, LD6:6, LD3:3andLD2:2, but not in DD and LD45 min:45 min (Figure 3). Among these strains, *wc-1^RIP^* is a *wc-1*-disrupted strain using the RIPing method [23]; *wc-2^KO^* and *frq^10^* are deletion mutants lacking functional WC-2 and FRQ, respectively; *wcc^DKO^* is a double knockout strain of *wc-1* and *wc-2*. The strains were exposed to a light intensity of 5000 lux, during the light periods. These data suggest that under short LD conditions, the conidiation rhythms cycle to the environment even without a functional circadian clock. We have also shown that in WT, the FRQ protein oscillation is masked under LD cycles (6 ≤ *T*< 24), but free runs when *T* < 6 (Figure 2). Taken together, these results suggest that the FWO system is not required for the conidiation banding rhythms driven by the cycling environment.

#### 3.1.4. Conidiation Rhythms of Photosensor-Related Mutants under Short LD Regimes

First, we investigated the impacts of individual photosensing-associated genes on the conidiation rhythms under short LD cycles in Δ*cry*; Δ*vvd*; Δ*phy-1*; Δ*phy-2*; Δ*nop-1* and in *cog-1*, a newly identified mutant showing circadian conidiation rhythmicity under constant light [14]. WC-1, VVD and CRY are blue light sensors while PHY-1 and PHY-2 are potential red-light receptors [24,27,35]. Unexpectedly, all of these strains showed conidiation rhythms with periods in accordance with the LD cycling periods, under conditions of LD12:12, LD6:6, LD3:3 and LD2:2 (Figure 4A–D). Together with the fact that the *wc-1**^RIP^* strain can be entrained at LD2:2, these results suggest that either there exists additional photosensor(s), or the lack of an individual photosensor is not sufficient to abolish the adaptation of conidiation to short LD cycling cues.

With this in mind, we investigated the conidiation rhythms in strains bearing multiple mutations/deletions of light-sensor or related genes, which included Δ*cry*, Δ*vvd*; Δ*cry*, Δ*wc-1*; Δ*vvd*, *cog-1*; Δ*wc-1*, *cog-1*; Δ*vvd*, Δ*wc-1*, *cog-1*; Δ*cry*, Δ*vvd*, *cog-1*; Δ*cry*, Δ*vvd*, Δ*wc-1*; Δ*cry*, Δ*wc-1*, *cog-1*, Δ*cry*, Δ*vvd*, Δ*wc-1*, *cog-1* and WT as control. Unexpectedly again, all of these strains still showed conidiation rhythms in short LD cycles (Figure 4E,F and Appendix A).

We also observed these strains in the red light/dark (RD) and blue light/dark (BD) cycles. In BD6:6 (blue light intensity: 5000 lux), all these strains showed conidiation rhythms (Appendix A). In BD3:3 and BD2:2 (blue light intensity: 5000 lux), all strains except *fwd-1*^RIP^ showed conidiation rhythms (Appendix A). However, in BD3:3 (blue light intensity: 1000 lux), many strains including *wcc**^DKO^*; *wc-1**^RIP^*; *fwd-1**^RIP^*; Δ*wc-2*; *frq**^10^*; Δ*cry*, Δ*wc-1*; Δ*cry*, Δ*vvd*, Δ*wc-1* and Δ*cry*, Δ*vvd*, Δ*wc-1*, *cog-1* showed no conidiation rhythms (Appendix A). The *fwd-1^RIP^* strain showed conidiation rhythms only in BD6:6 (blue light intensity: 5000 lux) but not in BD3:3 (blue light intensity: 5000 lux and 1000 lux) and BD2:2 (blue light intensity: 5000 lux) (Appendix A). In contrast, in the RD cycles, all clock gene mutation strains showed no entrained rhythms in RD6:6, RD3:3 and RD2:2 (red light intensity: 5000 lux), and *fwd-1^RIP^* showed conidiation rhythm under BD6:6 but not under BD3:3 and BD2:2 when the blue light intensity was 5000 lux or 1000 lux (Appendix A). Red light can also be sensed by some fungi, e.g., *Aspergillus nidulans*, *Trichoderma atroviride* and *Neurospora*, which is important for the asexual/sexual transition during development, hyphal growth and DNA stability [24,27,35]. Illumination of high-intensity blue light causes self-oxidative damage and aggregation of VVD [48]. Here we show that both the WT and *vvd**^KO^* strain were entrained to certain BD cycles under 1000 lux and 5000 lux (Appendix A), suggesting that this characteristics of VVD does not significantly contribute to the response to blue light cycles. Therefore, these data suggest that the conidiation rhythmicity is controlled predominantly by the blue light sensors and not the red-light sensors.

The temperature increase that accompanied the switch-on of the lighting in the incubator, which might also have affected the conidiation. To rule out this possibility, we measured conidiation rhythms under temperature cycles (25.5 °C 6 h: 24.5 °C 6 h) as the range of incubator temperature is ~±0.5 °C in our experiments, as previously described [14]. The results showed no resonance in all tested strains (Appendix A), demonstrating this temperature variation is not sufficient for entrainment. Taken together, these results show that all of these tested light sensors and associated factors were not critical for the conidiation rhythms. Instead, an unidentified blue light photoreceptor might be involved in controlling the conidiation rhythms.

The effects of different photosensing-associated genes on the production of carotenoid in all of these strains in constant light were also analyzed. As has been previously shown, the strains containing *vvd* mutations, including Δ*vvd*; Δ*vvd*, *cog-1*; Δ*cry*, Δ*vvd* and Δ*cry*, Δ*vvd*, *cog-1* displayed bright orange color in their mycelia [14,32]. In contrast, the mycelia color of strains Δ*vvd*, Δ*wc-1*, *cog-1*; Δ*cry*, Δ*vvd*, Δ*wc-1* and Δ*vvd*, Δ*cry*, Δ*wc-1*, *cog-1* was not bright orange (Appendix A). These data suggest that some of these light sensors might regulate the functions of others.

In contrast to our results, Nsa et al. showed that the Δ*wc-1* strain exhibit no obvious rhythms under a number of T-cycle conditions including LD6:6, LD9:9, LD12:12 and LD14:14 [14]. However, we used a white light intensity (~5000 lux) that was much higher than that used by Nsa et al. (1200 lux), which may account for the inconsistency. In support of this view, when we grew the clock mutants in LD conditions and the light intensity was approximately 1000 lux, and conidiation rhythmicity was abolished at the tested LD conditions (Figure 5A,B, Appendix A). Conversely, the conidiation rhythms in clock gene mutants, *wc-1^RIP^*, *wcc^DKO^*, *frq^10^* and Δ*wc-2*, were present in 5000 lux blue light but were absent in 1000 lux (Appendix A). These data suggest that while clock components might act to promote the function of a potential photoreceptor to elicit conidiation rhythms, the response of clock mutants to a high-intensity but not low-intensity light, suggests that higher-intensity light might overcome the repression effect caused by the loss of clock components (Figure 5C–E).

It is intriguing to note that in BD3:3 (blue light intensity: 1000 lux), the strains *cog-1*; Δ*wc-1*, *cog-1* and Δ*vvd*, Δ*wc-1*, *cog-1* exhibited biomodal patterns on the first day (Appendix A), reflecting the role of the CDO pathway in mediating the resonance of conidiation rhythmicity although the underlying mechanisms remain unclear.

#### 3.1.5. Implication of FWD-1 in Regulating the Conidiation Rhythms under Short LD Cycles

To probe the possible mechanisms regulating the adaptation to LD cycles, we also checked the conidiation rhythms in the *fwd-1^RIP^* strain that lacks the functional *fwd-1* gene, as FWD-1 has been implicated in *Neurospora* circadian clock through mediating the FRQ ubiquitination and turn over. *Neurospora* FWD-1 is the homolog of *Drosophila* Slimb, a ubiquitin E3 ligase [39,49].

The *fwd-1^RIP^* strain exhibited no overt conidiation bands under LD3:3 and shorter LD cycles (Figure 4, Figure 5 and Appendix A). The mutants lacking tested photosensors showed normal responses to short LD cycles, suggesting that the phenotype of *fwd-1^RIP^* is not based on the impacts of FWD-1 on these genes, i.e., additional photosensor(s) controlled by FWD-1 might be implicated in regulating the conidiation rhythms under short LD cycles. Alternatively, a potential factor connecting the specific photosensor to conidiation and, that is controlled by FWD-1, might be implicated (Figure 5C,D).

#### 3.1.6. Transcriptomic Analysis of Genes Driven by High-Intensity Light

The above data suggest that high-intensity light influences conidiation rhythms, thus we conducted RNA-seq analysis to assess the changes of gene expression in response to high-intensity light. We compared the transcriptomic changes of the strain Δ*cry*, Δ*vvd*, Δ*wc-1*, which contains no known blue light photoreceptors, after exposure for 45 min to either 1000 lux or 5000 lux white light (Figure 6A). The results showed that upon light exposure, a variety of genes involved in metabolism and gene expression were up-regulated (folds ≥ 1.5). For the genes exclusively driven by 5000-lux light, one putative motif (GAXGA) was identified by the XXmotif online tool (http://xxmotif.genzentrum.lmu.de/, accessed on 15 January 2018), which is present in 56 promoter areas of the 73 genes (Figure 6B,C). Light of 5000-lux driven the exclusive expression of 73 specific genes, many of which were enriched in fatty acid metabolism, ribosome biogenesis, etc. (Figure 6D; Appendix A). The induced expression of three representative genes (NCU00298, NCU00992 and NCU09771) were confirmed in Δ*cry*, Δ*vvd*, Δ*wc-1*; Δ*phy-1*, Δ*phy-2* and Δ*nop-1* by qRT-PCR (Figure 6E–G). Although the pathways of ribosome biogenesis and propanoate metabolism were induced in both 1000-lux and 5000-lux light (Appendix A), the genes involved were different, suggesting that higher-intensity light plays specific roles in regulating the expression certain genes and metabolism.

Taken together, these data confirm that high-intensity light induces the expression of a set of genes in clock mutants that might represent the targets for studying the potential unidentified photoreceptor(s).

## 4. Discussion

Within a certain range, the cyclical environmental changes (*T*-cycles) can drive rhythm with the same periodicity as the environment even when these are non-24-h cycles, which mean *T*-cycles mask the endogenous circadian periodicity. This range varies in different organisms [50,51]. Usually, entrainment involves the synchronization of the innate circadian clock to the environmental cues. It is important to note that entrainment here means a rhythm displaying a period that equals the length of the environmental cycle and having a stable phase angle, which takes no account of whether the innate circadian system is synchronized or not [9,14]. In *Neurospora*, we show that under almost all of the short LD cycles, including even very short cycles, such as LD1:1, the conidiation rhythms were driven by the *T*-cycles. The conidiation rhythms under short *T*-cycles are not autonomous as they disappeared in constant dark; instead, they are the hourglass-type rhythms.

Similar to the conidiation rhythms, the period of FRQ changed in accordance with the LD cycles when *T* > LD3:3. In contrast, under LD cycles with *T* ≤ LD3:3, FRQ level no longer oscillated with the T cycles, and instead exhibit periods close to 24 h, i.e., the masking effects faded while the FRQ rhythm free ran. The dynamic modeling results showed the superimposition of endogenous and the light-driven rhythms, which can also be observed in the western blot results of FRQ protein under very short LD cycles (*T*≤ LD3:3) (Figure 2). The inconsistency between conidiation rhythms and FRQ rhythms might reflect the decoupling between the oscillator and the clock-controlled output processes when *T* < LD3:3.

In many species, it has been demonstrated that non-24 h *T*-cycles decrease their fitness or adaptation [18], suggesting the adaptation of the endogenous rhythm of an organism to external cycling cues is critical for the development and growth. We show that the growth rate of *Neurospora* was slower under LD12:12 compared to many other conditions. However, the proportion of microconidia produced in LD12:12 prevailed over those in other conditions.

Faster growth of an organism does not necessarily mean optimal adaptation to the environment [18]. Instead, the capability to produce variable and fertile progenies may be more crucial. *Neurospora* microconidia function predominantly as spermatid [47], suggesting that LD12:12 might be optimal for sexual reproduction. It is likely that in nature, the circadian clock renders *Neurospora* better prepared to use its sexual reproduction strategies to cope with constantly changing environmental stresses. Compared to the findings in some other organisms, these results reflect that different organisms may employ different strategies for adaptation and propagation [52]. In addition, the faster growth in non-24 h conditions might help *Neurospora* escape from the abnormal environment.

In *Neurospora*, it has been claimed that WCC and VVD are responsible for most light-dependent physiological processes, if not all. Surprisingly, we show that upon removal of several photosensors, the conidiation rhythms still exist in short LD cycles. The clock mutants showed resonance to 5000 lux white light but not 1000 lux white light [14] (Figure 3 and Figure 5). Moreover, the conidiation rhythmicity only occurs in blue light but not red light (Appendix A). These data suggest the presence of unidentified blue photoreceptor(s) which drives conidiation rhythmicity. The conidiation rhythms in short LD cycles are independent of either FWO or CDO, though CDO and FWD-1 are involved in modulating the conidiation rhythms in short LD cycles.

Under 1000 lux white light, clock mutants displayed no conidiation rhythms, suggesting that the circadian clock functions to maintain or promote conidiation responses to short LD cycles. These data also suggest that the circadian clock might help *Neurospora* to promptly adjust its conidiation according to changeable light condition in the natural environment; for instance, caused by cloud movement or shadows being cast in the daytime. Under high-intensity light, even the clock mutants showed conidiation rhythms, suggesting the unidentified photoreceptor(s) might only sense very strong light, which can confer the condition rhythmicity in clock mutants.

The light characteristics including wavelength and intensity affects fungal germination, growth, development and metabolism and light initiates adaptation in a comprehensive set of metabolic pathways and [53,54,55]. In this work, we showed that light intensity induces differential gene expression at the transcriptomic level and found that a subset of metabolic pathways was affected upon higher light exposure relative to lower light, suggesting that additional photoreceptor(s) than those that are known (e.g., WC-1, VVD and CRY) play a role in regulating light-induced metabolic adaptation (Figure 6).

Taken together, these results provide insights into the resonance and adaptation of *Neurospora* circadian clock and conidiation rhythms to short LD cycles, which may benefit the understandings of the basic regulatory aspects of circadian clock. In addition, these findings might help us understand the changes and underlying mechanisms in circadian rhythms under some special conditions, e.g., the organisms living in the space laboratory exposed to the very short LD cycles (90–120-min), or the welfare of the people who are required to live and work under non-24 h work/rest schedules, e.g., maritime operations, oil mining and some other industries [56,57].

## Figures and Tables

**Figure 1 jof-08-00027-f001:**
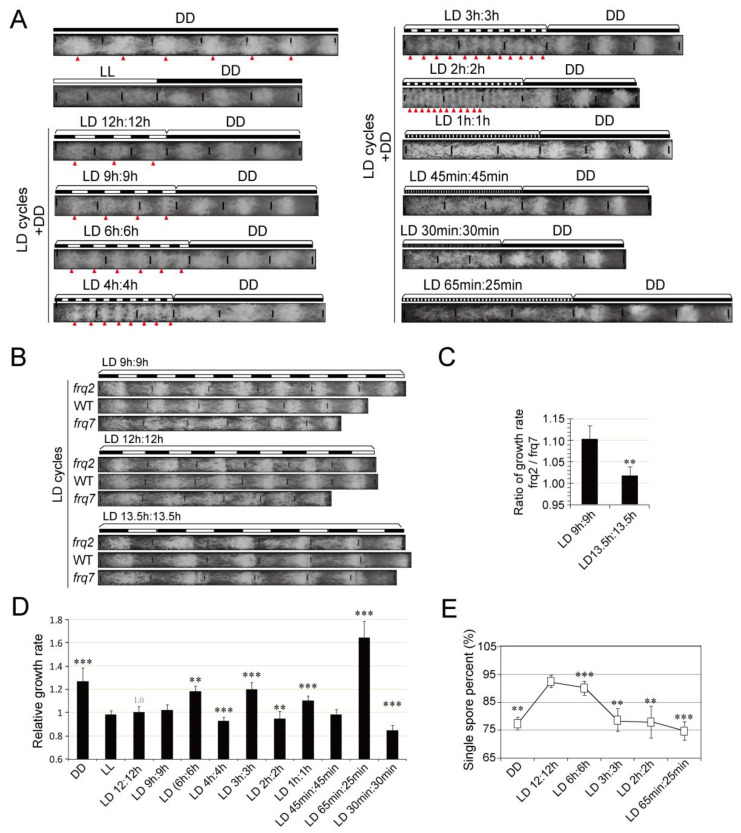
Conidiation rhythms, growth, and adaptation of *Neurospora* in short LD cycles. (**A**) Conidiation rhythms of *Neurospora* under a series of LD cycles. The black and white bars denote different LD regimes. Triangles denote the conidiation bands, (**B**) growth of *Neurospora* strains in race tubes under a series of LD cycles. Data are mean ± SD, *n* = 3. Representative results are shown (*n* ≥ 3), (**C**) ratio of the growth rates between *frq^2^* and *frq^7^* calculated according to the results of (**B**). Data are mean ± SD, *n* = 3, (**D**) relative growth rate of WT strain calculated according to the results of (**B**). The daily growth lengths were normalized to the data in LD12:12. Data are mean ± SE, *n* = 3, (**E**) proportion of microconidia produced under different LD cycles. Data are mean ± SE, *n* = 3. The light intensity was 5000 lux. ** represents *p* ≤ 0.01 and *** represents *p* ≤ 0.001.

**Figure 2 jof-08-00027-f002:**
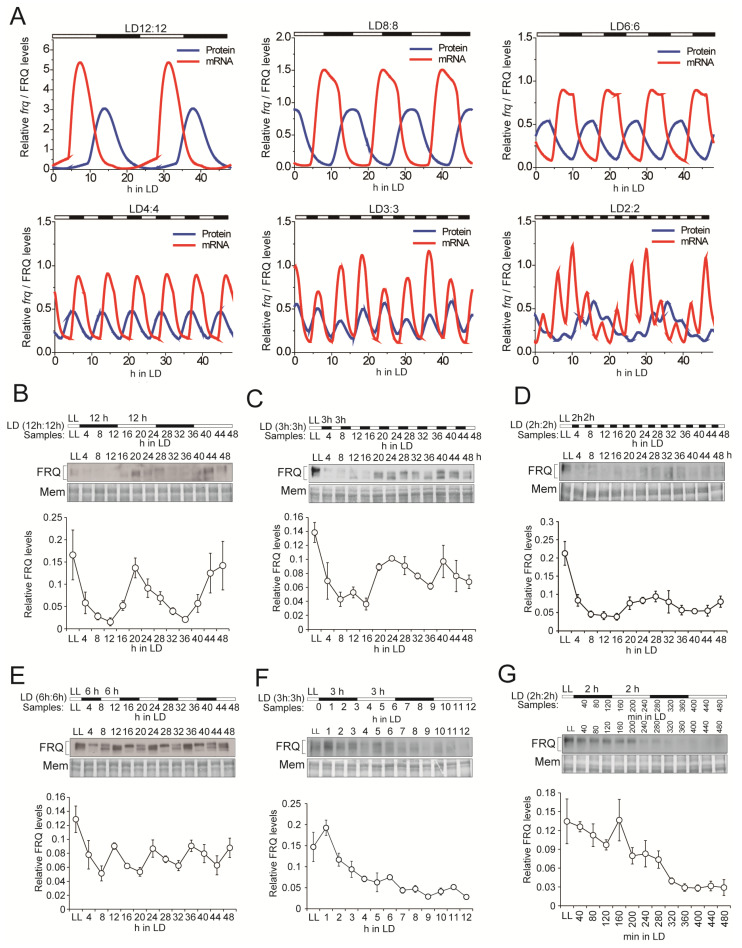
The FRQ protein levels under different LD routines. (**A**) Dynamic modeling results showing the time series of *frq* mRNA and FRQ protein under different light-dark cycles. Both the *frq* mRNA and FRQ protein can be entrained to LD12:12, LD8:8, LD6:6 and LD4:4. As the frequency of the light signal increases to LD3:3, the *frq* mRNA and FRQ protein still exhibit the oscillation with a period of approximately 6 h. If the frequency of the light signal is further increased to LD2:2, although the *frq* mRNA can respond to the induction by the light, the FRQ levels show the free running pattern with some fluctuations, (**B**–**G**) Western blot results showing FRQ levels in different LD cycles. The LD routines are LD12:12 for two cycles (**B**), LD3:3 for 48 h (**C**), LD2:2 for 48 h (**D**), LD6:6 for 48 h (**E**), LD3:3 for two cycles (**F**) and LD2:2 for two cycles (**G**), respectively. The black and white bars denote different LD regimes. Membranes (Mem) stained with Amido Black were used as loading control. The densitometric analysis of FRQ levels is shown at the bottom of each panel. Representative data of triplicates are shown. The light intensity was 5000 lux. Data are mean ± SE, *n* = 3.

**Figure 3 jof-08-00027-f003:**
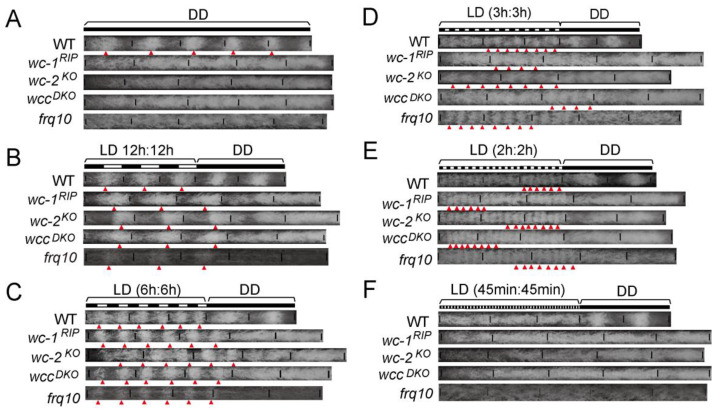
The conidiation rhythms of neurospora strains in short LD cycles. The conditions were DD (**A**), LD12:12 (**B**), LD6:6 (**C**), LD3:3 (**D**), LD2:2 (**E**) and LD45 min:45 min (**F**). Representative results (*n* ≥ 3) are shown. Triangles denote the conidiation bands. The strains were grown under white light and the light intensity was 5000 lux.

**Figure 4 jof-08-00027-f004:**
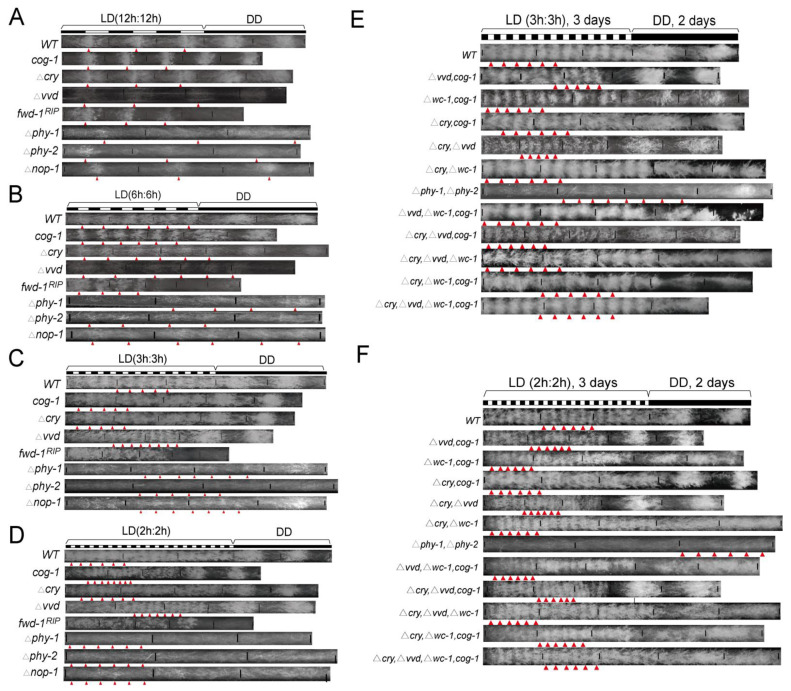
Impact of light sensors on the adaptation to short LD cycles. Race tube results of indicated strains under indicated LD cycles (**A**–**F**). Represent results (*n* ≥ 3) are shown. Triangles denote the conidiation bands. The strains were grown under white light and the light intensity was 5000 lux.

**Figure 5 jof-08-00027-f005:**
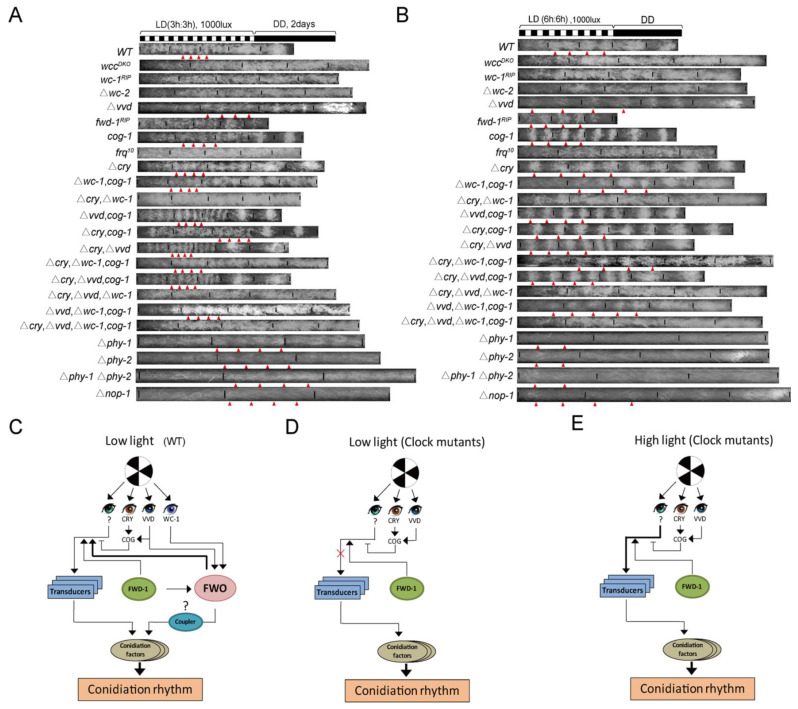
Conidiation rhythms in low white light (1000 lux). (**A**,**B**) Race tube results of conidiation rhythms of indicated strains in LD3:3 (**A**) and LD6:6 (**B**). Representative results are shown (*n* ≥ 3). The white light intensity was 1000 lux. Triangles denote the conidiation bands. (**C**–**E**). Schematics of the control of conidiation rhythms under short LD cycles of WT in low-intensity light (**C**) and clock mutants in low-intensity light (**D**) and high-intensity light (**E**). WC-1, VVD and additional photoreceptor(s)are implicated in the regulation of the conidiation rhythms. In 6 h≤ *T* ≤ 24 h, the FWO system is entrained, and the FRQ rhythms are not endogenous as they cannot be maintained in constant dark. The question masks denote unidentified photoreceptor and putative coupler linking FWO and conidiation, respectively. Transducers denote the factors linking photoreceptor and conidiation.

**Figure 6 jof-08-00027-f006:**
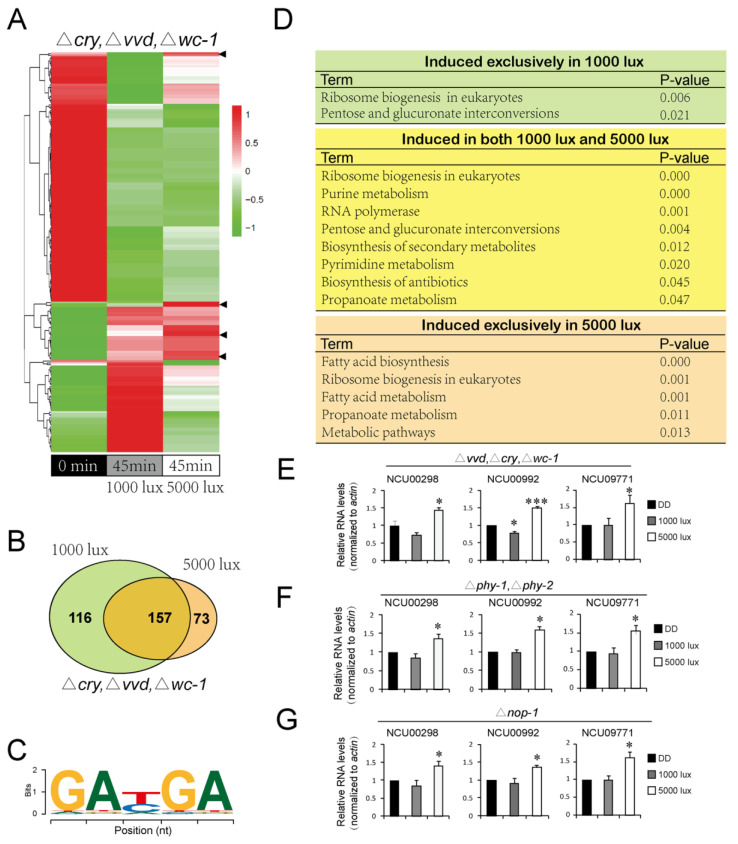
Characterization of genes showing light responses in high-intensity white light. (**A**) Heat map of the differentially expressed genes in Δ*vvd*, Δ*cry*, Δ*wc-1* at DD24, 1000-lux white light and 5000-lux white light. “0 min” refers to the sampling time prior to light exposure, “45 min” refers to the sampling time point of 45 min after light exposure. The heat map was created according to the RNA-seq data, (**B**) Venn diagram showing the genes up-regulated in 1000 lux and 5000 lux relative to DD24, respectively, (**C**) sequence logo of a predicted motif, (**D**) distribution of affected KEGG pathways according to the up-regulated genes revealed from RNA-seq, (**E**–**G**) qRT-PCR validation of three genes specifically induced by 5000-lux light in Δ*vvd*, Δ*cry*, Δ*wc-1* (**E**), Δ*phy-1*, Δ*phy-2* (**F**) and Δ*nop-1* (**G**) strains. The gene expression was normalized to *actin*. The levels at DD24 were normalized to 1.0. Data are mean ± SE, *n* = 3. * represents *p* ≤ 0.05 and *** represents *p* ≤ 0.001.

## Data Availability

RNA-seq data sets are available at the GEO database (GSE108814).

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
