# Peer review of "The Resonance and Adaptation of Neurospora crassa Circadian and Conidiation Rhyth ms to Short Light-Dark Cycles"

_jof, 2021, doi:10.3390/jof8010027_

Round 1

Reviewer 1 Report

Hereafter you will find the major comment about this article. The scientific question assessed by the authors, is undoubtedly interesting when working on controlled reactors and more over very pertinent when applying such approach at production level (scale up). This work is consistent on its own and very well described. It should be proposed as it for publication.

Author Response

Although there are no specific comments, we also made revisions to improve the manuscript as below:

  1. The affiliation of the second author Luyao Li should be “1” instead of “2”.
  2. At the end of the first sentence in the second paragraph in page 8, we added “and WT as control”.
  3. Page 4, at the end of the third sentence in the first paragraph, the right bracket was removed.
  4. In supplemental Figure S5, the labeling of 5D was “RD6:6” which is incorrect. We have changed it to be “BD6:6”. And in accordance to their appearance in the text, the order of the panels in Figure S5 has been modified and the citations of Figure S5 in the text have been updated.
  5. In the last paragraph in pp 9, the original sentence of “It is intriguing to note that in BD cycles” has been changed to be “It is intriguing to note that in BD3:3 (blue light intensity: 1000 lux)” to make it clearer. And the cited figure has been changed to be Figure S5A due to the modification described in 8.

Reviewer 2 Report

This work investigates the response of the Neurospora circadian clock and one of its output rhythms, conidiation, to very short light/dark cycles (symmetrical T-cycles with less than 6 hr duration).  The results suggest that conidiation directly responds to light signals resulting in apparent rhythmicity even under LD1:1 cycles, whereas the molecular clock (assessed by FRQ protein levels) stops running below LD3:3. The data indicate the presence of a yet unknown blue light receptor, which is probably strongly activated by intense blue light, but its function may be affected by clock components at lower light levels. The genome-wide effect of lower vs higher intensity light on gene expression in a strain lacking the three main known light receptors was tested by RNA-seq analysis. Differentially expressed genes from this experiment could represent the potential targets of the unknown receptor.

I think this is a nice set of results, which deserves further analysis and experimental work to have a really strong impact. However, the work is publishable in its present work after considering a few stylistic and conceptual issues:

  1. Lines 273-277

This section describes how conidiation of the different mutants and mutant combinations respond to monochromatic T-cycles. The text here is extremely confusing and makes it very hard to get to the point. Please rewrite these sentences and make it absolutely clear which strain shows conidiation rhythm under which T-cycle.

  1. T-cycles do not “induce” rhythms. T-cycles (and in general: environmental cycles) drive rhythms within the particular range of entrainment.

  1. Lines 384-385

I would say that conidiation rhythms under very short T-cycles are not AUTONOMOUS.

  1. Line 394

I think it is better to say that under very short T-cycles conidiation, as a normally clock-controlled output process (and not an extrinsic rhythm) is decoupled from the oscillator.
